# Unraveling the Complex Molecular Interplay and Vascular Adaptive Changes in Hypertension-Induced Kidney Disease

**DOI:** 10.3390/biomedicines12081723

**Published:** 2024-08-01

**Authors:** Lyubomir Gaydarski, Iva N. Dimitrova, Stancho Stanchev, Alexandar Iliev, Georgi Kotov, Vidin Kirkov, Nikola Stamenov, Tihomir Dikov, Georgi P. Georgiev, Boycho Landzhov

**Affiliations:** 1Department of Anatomy, Histology and Embryology, Medical University of Sofia, 1431 Sofia, Bulgaria; lgaidarsky@gmail.com (L.G.); stanchev_1989@abv.bg (S.S.); dralexiliev@abv.bg (A.I.); nstamenov@medfac.mu-sofia.bg (N.S.); landzhov_medac@abv.bg (B.L.); 2Department of Cardiology, University Hospital “St. Ekaterina”, Medical University of Sofia, 1431 Sofia, Bulgaria; dimytrova@yahoo.com; 3Department of Rheumatology, Clinic of Rheumatology, University Hospital “St. Ivan Rilski”, Medical Faculty, Medical University of Sofia, 1612 Sofia, Bulgaria; gn_kotov@abv.bg; 4Department of Health Policy and Management, Faculty of Public Health “Prof. Dr. Tzekomir Vodenicharov”, Medical University of Sofia, 1431 Sofia, Bulgaria; vidinkk@abv.bg; 5Department of General and Clinical Pathology, Medical University of Sofia, 1431 Sofia, Bulgaria; tdikov@medfac.mu-sofia.bg; 6Department of Orthopedics and Traumatology, University Hospital Queen Giovanna-ISUL, Medical University of Sofia, 1527 Sofia, Bulgaria

**Keywords:** angiogenesis, vascular adaptive mechanisms, arterial hypertension, apelin receptor, neuronal nitric oxide synthase, vascular endothelial growth factor

## Abstract

Angiogenesis, the natural mechanism by which fresh blood vessels develop from preexisting ones, is altered in arterial hypertension (AH), impacting renal function. Studies have shown that hypertension-induced renal damage involves changes in capillary density (CD), indicating alterations in vascularization. We aimed to elucidate the role of the apelin receptor (APLNR), neuronal nitric oxide synthase (nNOS), and vascular endothelial growth factor (VEGF) in hypertension-induced renal damage. We used two groups of spontaneously hypertensive rats aged 6 and 12 months, representing different stages of AH, and compared them to age-matched normotensive controls. The kidney tissue samples were prepared through a well-established protocol. All data analysis was conducted with a dedicated software program. APLNR was localized in tubular epithelial cells and the endothelial cells of the glomeruli, with higher expression in older SHRs. The localization of nNOS and VEGF was similar. The expression of APLNR and nNOS increased with AH progression, while VEGF levels decreased. CD was lower in young SHRs compared to controls and decreased significantly in older SHRs in comparison to age-matched controls. Our statistical analysis revealed significant differences in molecule expression between age groups and varying correlations between the expression of the three molecules and CD.

## 1. Introduction

Angiogenesis is the natural mechanism by which fresh blood vessels develop from preexisting ones, serving a pivotal function in diverse biological processes. It is crucial for sustained primary tumor expansion, wound recovery, and the generation of granulation tissue [1]. Angiogenesis encompasses the expansion of blood vessels from preexisting vasculature and represents a critical phase in the progression of tumors from a benign to a malignant state. This mechanism is governed by several components, including vascular endothelial growth factor (VEGF), which primarily prompts the formation of new blood vessels through processes such as sprouting and splitting [1,2]. Capillary density (CD), vessel lumen area, and vessel lumen perimeter are effective morphometric parameters used to assess angiogenesis [3].

Arterial hypertension (AH) is a chronic medical condition and a leading factor in cardiovascular and kidney injury, including chronic kidney disease (CKD). It represents a significant healthcare and economic challenge in aging populations [4]. Although the precise causes of hypertension remain partly unclear, its onset is associated with various factors [5]. The renal damage caused by hypertension is a complex process that significantly contributes to CKD progression, which is closely intertwined with hypertension, acting as both a contributor to and a consequence of elevated blood pressure (BP) levels [6]. As kidney function declines, BP tends to rise, resulting in hypertension in over 85% of CKD patients [6]. Therefore, maintaining normal BP levels in CKD is essential for slowing kidney deterioration and reducing cardiovascular risks [7]. Various histomorphometric parameters are used to assess the severity of hypertension-induced renal damage [8,9,10]. One such parameter is CD. The correlation between decreased renal CD and hypertension is well documented in the literature [11,12]. Studies indicate that capillary rarefaction, characterized by decreased vascular density, plays a significant role in CKD progression and hypertension-induced renal damage. Peritubular capillary rarefaction, commonly observed in conditions like hypertensive nephrosclerosis, diabetic nephropathy, and CKD, is associated with impaired kidney function and structural alterations in the renal microvasculature [11,12,13]. Additionally, literature data underscore the significance of capillary rarefaction as a marker of essential hypertension [13]. These findings point to the crucial role of CD and capillary rarefaction in hypertension-induced renal damage, highlighting the importance of vascular changes in understanding and managing hypertensive kidney disease.

The apelin system involves the apelin receptor (APLNR), a G protein-coupled receptor, and its endogenous ligands, apelin and elabela. Although APLNR shares approximately 50% homology with the type I angiotensin receptor, it does not interact with angiotensin II [14]. With respect to the kidney, the apelin system has a potential role in the maintenance of renal functions under physiological and pathological conditions. Apelin promotes vasodilatation of the afferent and efferent glomerular arterioles via nitric oxide (NO)-dependent mechanism, thus opposing the effects of the renin–angiotensin system [15]. It seems that the apelinergic system pathways are functionally closely associated with the activity of the nitric oxide synthase (NOS) pathway [16]. In fact, the administration of L-NAME, an inhibitor of the isoforms of NOS, leads to decreased renal expression of apelin and APLNR [17]. On the other hand, in the case of vascular damage, apelin can serve as a vasoconstrictor [18]. In addition, apelin attenuates renal inflammatory and fibrotic changes as well as renal tubular damage [16,19]. Elabela is mainly expressed in the kidney and shows a pronounced renoprotective role as it attenuates the development of hypertensive nephropathy, including glomerular and tubulointerstitial injury [20]. Elabela has a pronounced role during the prenatal period of life as it is involved in processes such as cell differentiation, migration, and neovascularization [21].

Apelin plays a potential role in the maintenance of glomerular integrity as it diminishes the development of glomerular hypertrophy [22]. APLNR localization within the kidney shows heterogeneous distribution with more prominent expression within the renal cortex compared to the renal medulla [22]. APLNR expression is found in endothelial cells and vascular smooth muscle cells of the intrarenal vasculature, podocytes, cells of the juxtaglomerular apparatus, loop of Henle, distal tubules, and collecting ducts [22].

Under hypertensive conditions, APLNR is upregulated with a decrease in apelin expression [23]. The enhanced expression of APLNR provokes vascular smooth muscle cell proliferation [23]. Apelin-13 has a potential role in angiogenesis as it enhances the differentiation, migration, and proliferation of endothelial progenitor cells [24]. Elabela has a pivotal role in the maintenance of blood pressure, as its serum levels are decreased in the case of malignant hypertension and may serve as a marker of hypertension-induced renal damage [24]. On the other hand, Huang et al. suggested APLNR has no significant correlation with the risk of hypertension [25].

NO is a gaseous signaling molecule with pronounced antihypertensive and renoprotective effects. It is produced by three NOS isoforms, all expressed in the kidney. The distribution of neuronal NOS (nNOS) is confined mainly to Bowman’s capsule, macula densa, and inner medullary collecting ducts. Usually, the renal expression of inducible NOS (iNOS) is associated with underlying pathology. Endothelial NOS (eNOS) is predominantly found in the vascular endothelium [26]. NO has a pivotal role in the control of renal blood flow, glomerular hemodynamics, renin secretion, renal sympathetic innervation, and the tubuloglomerular feedback (TGF) mechanism [27,28]. TGF is an autoregulatory mechanism of the nephron controlling the glomerular arteriolar resistance. The increased delivery of fluid and sodium chloride concentration to the distal nephron detected by macula densa cells stimulates the secretion of renin, leading to vasoconstriction of the afferent arteriole via the renin–angiotensin system and provokes a decline in the glomerular filtration rate (GFR) [29]. The highest expression of NOS is found in the inner medulla [30]. The inhibition of nNOS affects VEGF response, leading to the development of hypertension as a result of pronounced vasoconstriction and sodium retention [31]. The selective inhibition of nNOS provokes glomerular injury, respectively, altered structure of the glomerular filtration barrier, and advanced renal damage in spontaneously hypertensive rats (SHRs), a classic experimental model of AH [32]. SHRs have an increased total NOS activity in the renal cortex compared to normotensive Wistar rats [33].

The VEGF family comprises VEGF-A, VEGF-B, VEGF-C, VEGF-D, and placental growth factor (PGF) in mammalian species, including humans [34]. VEGF-A has a pivotal role in angiogenesis, vascular permeability, endothelial cell survival, and migration [34]. VEGF-A mediates its effects by binding to its receptors, namely, VEGFR1 and VEGFR2 [35]. Within the kidney, VEGF-A is expressed in glomerular podocytes, epithelial cells of the distal tubules, and collecting ducts [36,37]. It seems VEGF-A expression within the distal tubular segments of the nephrons is weaker compared to the podocytes [38]. Kang et al. described a correlation between the peritubular CD and tubular VEGF expression [39]. Recent research suggests a potential link between VEGF and hypertension-induced renal injury [40,41]. VEGF’s involvement in hypertension development is well documented, as it influences the structural and functional aspects of the vascular system. During the initial phases of hypertension, VEGF is believed to promote blood vessel remodeling and impair vascular function [41]. As hypertension progresses, VEGF’s role becomes more detrimental. Elevated VEGF levels have been linked to heightened blood vessel permeability, potentially resulting in the leakage of plasma proteins into the interstitial space. This phenomenon contributes to vascular damage and further exacerbates the progression hypertensive damage [42].

Based on the above, there appear to exist intricate interactions within the complex signaling pathways involving the apelinergic system, nNOS, and VEGF. Building upon these findings, we hypothesize that alterations in APLNR expression and, indirectly, the apelinergic system might play a crucial role in regulating the signaling of nNOS and VEGF in the context of hypertension-induced depletion of renal angiogenesis. Therefore, the aim of our study was to elucidate the complex interactions between the apelinergic system and the expression of nNOS and VEGF during the onset and progression of hypertension-induced renal injury in an experimental model of AH. To do so, we evaluated the immunohistochemical expression of the APLNR, nNOS, and VEGF and correlated it with CD, which serves as a reliable quantitative parameter for assessing angiogenesis [3] and hypertension-induced renal damage [11,12,13].

## 2. Materials and Methods

### 2.1. Experimental Animals

In the current study, we employed two different age groups of SHRs: 6-month-old (representing early-stage hypertension) and 12-month-old (late-stage hypertension) [43]. As controls, we used age-matched normotensive Wistar rats (WRs). Each group comprised three male rats randomly selected from a larger population of SHRs and WRs, respectively, from the Laboratory of the Department of Anatomy, Histology, and Embryology at the Medical University of Sofia, Bulgaria. All animal procedures followed previously established protocols [44]. Systolic and diastolic arterial blood pressure were measured using the tail cuff method with a Model MK-2000ST (manufactured by Muromachi Kikai Co., Ltd., Tokyo, Japan).

### 2.2. Tissue Preparation

The anesthesia was administrated intraperitoneally with Thiopental at a dose of 40 mg/kg body weight. Afterward, their chest cavities were opened, and transcardial perfusion was conducted using a 4% paraformaldehyde solution. The kidneys were quickly removed and immersed in a 10% neutral phosphate-buffered formalin solution for at least 24 h to fix them. Following a standardized procedure [45], the kidneys were sliced parallel to their longitudinal axis. Then, the kidney samples were dehydrated using alcohol solutions with gradually increasing concentrations, cleared with xylene, and embedded in paraffin.

### 2.3. Morphometric Analysis of CD

Slides containing kidney tissue samples were prepared for standard light microscopic examination as described previously [46]. Quantitative data were collected using the computerized image analysis system, NIS-Elements Advanced Research (Ver. 2.30). CD quantification was performed following well-established protocols [47,48].

### 2.4. Immunohistochemistry

The immunohistochemical analysis was conducted using the heat-induced epitope retrieval (HIER) technique, following established procedures [46]. We used a mouse monoclonal anti-APLNR IgG antibody (Santa Cruz Biotechnology Catalogue No. sc-517300, Santa Cruz Biotechnology, Inc., Heidelberg, Germany) at a concentration of 1:250, and a mouse monoclonal anti-nNOS IgG antibody (Santa Cruz Biotechnology Catalogue No. sc-398843) at a concentration of 1:500. Additionally, a mouse monoclonal anti-VEGF-A IgG antibody (Santa Cruz Biotechnology Catalogue No. sc-7269) at a concentration of 1:250 was used. All other steps followed the standardized protocol outlined in our previous research [41,46].

### 2.5. Semi-Quantitative Analysis

To conduct the semi-quantitative evaluation of the expression of APLNR, nNOS, and VEGF, we used ImageJ 1.52a software, which was sourced from the National Institute of Health (NIH) website (https://imagej.net/ij/). The staining intensity was assessed with the assistance of the IHC Profiler plugin, which was downloaded from the Sourceforge website (https://sourceforge.net/projects/ihcprofiler/, accessed on 30 July 2024) and followed a standardized procedure [Varghese]. The IHC Profiler categorized staining intensity into four levels: high positive (3+), positive (2+), low positive (1+), and negative (0). Our analysis included a minimum of ten randomly selected visual fields from each kidney. From each animal in each age group, we evaluated five slides obtained from each. The final score assigned to each RC and RM within a specific age group was calculated as the average score across all visual fields determined by the IHC Profiler.

### 2.6. Statistical Analysis

The statistical analysis was conducted using SPSS software v28.0.0.1 (IBM Corporation, Armonk, NY, USA). Data distribution was assessed for normality using the Kolmogorov–Smirnov test, revealing a non-normal distribution. Subsequently, the Mann–Whitney test was employed to evaluate statistically significant differences between CD at 6 and 12 months in both the RC and RM groups. Mann–Whitney U test was utilized to test for statistically significant difference between the immunohistochemical expression of the molecules in SHR and WR controls. Additionally, the analysis of variance (ANOVA) and post-hoc Tukey HSD tests were used to determine significant differences in the expression of the three molecules. Spearman’s correlation analysis was employed to investigate correlations between CD per slide and the expression of APLNR, nNOS, and VEGF, represented as the total score of the examined area on the slide. Furthermore, Spearman’s correlation test was conducted to assess the expression of the three molecules comprehensively. A standard significance level of α (*p*-value) = 0.05 was applied to all tests.

## 3. Results

### 3.1. Measurement of BP

The BP values of both age groups are presented in Table 1.

### 3.2. Immunohistochemical Expression and Semi-Quantitative Assessment

Immunoreactivity of the selected molecules was analyzed separately in the RC and RM. The results of the semi-quantitative analysis are summarized in Table 2.

#### 3.2.1. APLNR Expression

In the RC, APLNR expression was mainly visualized on the outer surface of the cell membrane of the epithelial cells of the proximal and distal contorted tubules, as well as the cell membrane of the epithelium of the parietal layer of the Bowman’s capsule, in both age groups. Weak APLNR immunoreactivity was registered in the glomeruli of 6-month-old animals. The expression of the APLNR was significantly higher in the RC of the 12-month-old animals. In the medulla, APLNR expression was mainly seen on the membrane of the epithelial cells of the collecting ducts and the loop of Henle (Figure 1).

The semi-quantitative analysis revealed that APLNR expression in the RC of 6-month-old SHRs was predominantly low positive (1+), with less than one-third of the slides showing negative (0) expression, compared to one-third of slides with low positive (1+) expression with the rest of slides exhibiting negative (0) expression in the RC of 6-month-old WRs (*p* = 0.007). In the RC of 12-month-old SHRs, for the APLNR, half of the slides had positive (2+) expression and some fields even displayed high positive (3+). In comparison, only 4% of the slides in the RC of the older control group demonstrated positive (2+) expression, as the vast majority of slides had low positive (1+) expression (*p* < 0.001). In the RM of young SHRs, half of fields had low positive (1+) immunoreactivity as compared to one third of the examined slides showing low positive (1+) expression in young WRs (*p* = 0.02). In old SHRs, nearly quarter of fields had high positive (3+) expression and the majority of slides had positive (2+) reactivity; a similar expression was noted in old WRs, where most slides were reported to have positive (2+) expression, yet no fields with high positive expression were detected (*p* = 0.03).

#### 3.2.2. nNOS Expression

The expression of nNOS in the RC was predominantly visualized on the cell membrane and in the cytoplasm of the epithelial cells of the proximal and distal ducts. In the glomeruli, immunoreactivity was detected mainly on the visceral layer of the Bowman’s capsule, with scant traces on the cell membranes of the parietal layer of the Bowman’s capsule. In the RM, the expression of nNOS was mostly located in the cytoplasm and the cell membrane of the epithelial cells of the collecting ducts and the loop of Henle (Figure 2).

The semi-quantitative analysis revealed that in the RC of 6-month-old SHRs, almost half of the assessed fields had low positive (1+) expression of nNOS, while the remaining 56% of fields displayed negative expression (0), whereas only 19% of fields displayed low positive (1+) expression in the control group. In the RC of older SHRs, the vast majority of fields displayed low positive (1+), with a few even showing positive (2+) expression, and only one-fifth of the fields had negative (0) immunoreactivity. These findings significantly contrasted to the 42% of fields with low positive (1+) expression in old WRs. In the medulla of young SHRs, most fields had negative (0) expression, with only 39% read as low positive (1+). Similar was the tendency in the control group. In the RM of 12-month-old animals, a similar result was registered, with 58% negative fields and 42% with low positive (1+) expression—a tendency mirrored in age-matched WRs.

#### 3.2.3. VEGF Expression

Immunohistochemical expression in the RC was mainly visualized within the cytoplasm of the epithelial cells lining the proximal and distal tubules, as well as in the visceral layer of the Bowman’s capsule in both age groups. It is to be noticed that the capillary tufts lacked any sign of VEGF expression in both 6- and 12-month-old animals. In the RM, immunoreactivity of VEGF was mainly registered in the cytoplasm of the epithelial cells of the collecting ducts and the loop of Henle (Figure 3).

VEGF expression was strongest in 6-month-old SHRs, where most slides showed positive (2+) expression across the RC, with one-fifth of the assessed fields having strong positive (3+) expression. In comparison, two-thirds of the slides displayed positive (2+) staining in the controls. Noticeable downregulation of VEGF expression was recorded in 12-month-old SHRs, as only 9% of fields had positive (2+) expression, half of the slides had low positive (1+) and 40% had negative (0) expression. The expression levels were similar in normotensive 12-month-old WRs. Similar results were reported in the RM of young SHRs, where nearly one-third of the fields had strong positive expression (3+), and 40% of the fields were positive (2+), which contrasted significantly with the control group. We registered a significant depletion of VEGF expression in the RM of 12-month-old SHRs, where most slides had mainly low positive (1+) expression, one-third of the slides were negative (0), and only 7% of the studied fields were positive (2+)—findings similar to those reported in the control group.

The results of the Mann–Whitney U test used to examine for significant differences between the immunoreactivity registered in SHRs and WRs are summarized in Table 3.

### 3.3. Analysis of CD

We performed the morphometric analysis of CD on H&E-stained slides (Figure 4). The highest CD was recorded in the RM of 6-month-old WRs, followed by the RC of 6-month-old WRs. In comparison CD was significantly lower in both RC and RM of 6-month-old SHRs. The lowest CD was recorded in the RC of 12-month-old SHRs, which proved to be significantly lower than the CD in the RC of age-matched controls. Intriguingly, the CD in the RC of old SHRs was nearly half that in young SHRs. While capillary rarefication was also observed in the controls, it was less pronounced. We reported a statistically significant difference between the CD in the RM of 12-month-old SHRs compared to controls of the same age group. Moreover, we registered a significant depletion of CD in the RM of 12-month-old animals compared to the CD of 6-month-old ones. The same tendency of diminishing CD was observed in 12-month-old WRs, but not as significant (Table 4 and Figure 5).

### 3.4. Statistical Assessment

#### 3.4.1. ANOVA and Post-Hoc Tukey HSD Tests

ANOVA performed on the expression levels of the three molecules in the RC of 6-month-old SHRs demonstrated a significant difference among group means (*p* < 0.001). Subsequent post-hoc Tukey’s HSD test confirmed significant pairwise differences between the expression of APLNR and VEGF, as well as between all three molecules (*p* < 0.05) Similar results were registered in the control groups, with the oly difference being that no significant disperity was registered between the expression of APLNR and nNOS (*p* = 0.3). In the RC of 12-month-old SHRs, the ANOVA results similarly indicated statistically significant difference in the expression of the molecules (*p* < 0.001), which was further supported by the post-hoc Tukey’s HSD test showing significant differences in pairwise comparisons between APLNR and nNOS and APLNR and VEGF (*p* < 0.001) with only nNOS and VEGF showing an insignificant difference in their expression (*p* = 0.8). Moreover, we reported significant differences between the expression of APLNR and VEGF, as well as between nNOS and VEGF (*p* < 0.001), but not between APLNR and nNOS (*p* > 0.05) in the RC of the control group of 12-month-old WRs.

In the RM of 6-month-old SHRs, we found statistically significant difference in molecule expression (*p* < 0.001). The post-hoc Tukey HSD comparisons demonstrated significant differences between the expression of APLNR and VEGF, as well as between nNOS and VEGF (*p* < 0.001). However, no significant difference was observed between APLNR and nNOS expression (*p* >0.05). Similar results were reported in the control group. In the RM of 12-month-old SHRs, ANOVA indicated a significant difference between the group means (*p* < 0.001). The post-hoc Tukey HSD comparisons revealed a non-significant difference between the expression of nNOS and VEGF (*p* = 0.6); however, APLNR displayed significant differences in its expression compared to nNOS and VEGF (*p* < 0.001). Similar was the tendency in the RM of 12-month-old WR controls. The results of the ANOVA and post-hoc Tukey’s HSD tests are summarized in Table 5.

#### 3.4.2. Correlations

In the RC of 6-month-old SHRs, significant correlations were found among molecular expressions and CD. APLNR showed a non-significant leak negative correlation with CD (r = −0.19, *p* = 0.3), indicating that decreased APLNR expression corresponds to increased CD, with approximately 4% of CD variability explained by APLNR (r^2^ = 0.04). While nNOS and CD exhibited a weak negative correlation that was not statistically significant (r = −0.25, *p* = 0.2), approximately 6% of CD variability was explained by nNOS (r^2^ = 0.06). Conversely, a significant moderate positive correlation was observed between VEGF and CD (r = 0.58, *p* = 0.0008), explaining around 34% of CD variability (r^2^ = 0.34). A weak negative correlation was found between APLNR and nNOS (r = −0.18, *p* = 0.6), explaining nearly 6% of the discrepancy between the molecules (r^2^ = 0.06). A moderate negative correlation was seen between APLNR and VEGF (r = −0.31, *p* = 0.04), explaining 10% of the difference. Similarly, a moderate negative correlation was observed between nNOS and VEGF (r = −0.38, *p* = 0.04), explaining 15% of the observed differences (r^2^ = 0.15). In the RC of age-matched controls, we report a weak negative correlation between APLNR and CD, which was not statistically significant (r = −0.35; *p* = 0.4), as only 13% of CD was related to APLNR expression (r^2^ =0.13). Similar was the correlation between nNOS and CD (r = −0.32; *p* = 0.04), and only 11% of CD was related to nNOS (r^2^ = 0.11). A moderate positive statistically significant correlation was observed between VEGF and CD (r = 0.39; *p* = 0.03), and 15% of CD was related to VEGF expression (r^2^ = 0.15). APLNR and nNOS exhibited a strong positive significant correlation (r = 0.22; *p* = 0.4), as 7% of their variability was inter-related (r^2^ = 0.07). Both APLNR (r = −0.44, r^2^ = 0.21)) and nNOS (r = −0.42, r^2^ = 0.19) had moderate negative significant correlations with VEGF (*p* < 0.05).

In the RC of 12-month-old SHRs, a moderate negative correlation was observed between APLNR and CD (r = −0.54, *p* = 0.002), with approximately 29% of the variability in CD explained by APLNR (r^2^ = 0.29). Conversely, a moderate positive correlation was found between nNOS and CD (r = 0.38, *p* = 0.04), explaining approximately 15% of CD variability (r^2^ = 0.15). Moreover, a strong positive correlation was detected between VEGF and CD (r = 0.60, *p* = 0.002), suggesting that approximately 38% of CD variability can be explained by VEGF (r^2^ = 0.38). A moderate negative correlation was found between APLNR and nNOS (r = −0.48, *p* = 0.01), indicating approximately 21% of the variability in nNOS dependent on APLNR (r^2^ = 0.21). Similarly, a moderate negative correlation was suggested between APLNR and VEGF (r = −0.41, *p* = 0.02), with approximately 18% of the variability in VEGF explained by APLNR (r^2^ = 0.18). A positive correlation was observed between nNOS and VEGF (r = 0.36, *p* = 0.04), indicating a moderate linear relationship that was statistically significant, with approximately 13% of the variability in VEGF explained by nNOS (r^2^ = 0.13). In the RC of 12-month-old controls we reported a weak negative insignificant correlation between APLNR and CD (r = −0.09; *p* = 0.6), as only 2% of CD could be explained by APLNR expression (r^2^ = 0.02). Similar was the relation between nNOS and CD (r = −0.22; *p* = 0.2), and only 5% of CD could be explained by nNOS expression (r^2^ = 0.05). On the other hand, VEGF and CD had a moderate positive significant correlation (r = 0.45; *p* = 0.01) and 20% of CD were dependent on VEGF (r^2^ = 0.20). APLNR and nNOS had a strong positive significant correlation (r = 0.93; *p* < 0.001), as 88% of APLNR expression could be explained by nNOS (r^2^ = 0.88). Both APLNR (r = −0.32) and nNOS (r = −0.30) had moderate negative significant correlations with VEGF (*p* < 0.05).

In the RM of 6-month-old SHRs, a weak negative correlation was observed between APLNR and CD (r = −0.24), but it was not statistically significant (*p* = 0.3). Approximately 5% of the variability in CD could be explained by the correlation with APLNR expression (r^2^ = 0.05). Similarly, a weak negative correlation was found between nNOS and CD (r = −0.31), not reaching statistical significance (*p* = 0.08), with approximately 9% of the variability in CD explained by nNOS (r^2^ = 0.09). In contrast, a strong positive correlation was observed between VEGF and CD (r = 0.66), highly statistically significant (*p* <0.001), explaining approximately 43% of the variability in CD (r^2^ = 0.43). Additionally, a strong positive correlation was found between APLNR and nNOS (r = 0.69), highly statistically significant (*p* < 0.001), explaining approximately 43% of the variability in nNOS (r^2^ = 0.43). A moderate negative correlation was observed between APLNR and VEGF (r = −0.54), statistically significant (*p* = 0.01), explaining approximately 26% of the variability in VEGF (r^2^ = 0.26). Moreover, a moderate negative correlation was found between nNOS and VEGF (r = −0.42), statistically significant (*p* = 0.02), explaining approximately 18% of the variability in VEGF (r^2^ = 0.18). Our findings in the RM of 6-month-old controls were similar, as a moderate negative correlation, statistically significant, was recorded between APLNR and CD (r = −0.31; *p* = 0.04), as 10% of CD was related to APLNR expression (r^2^ = 0.10). Similar was the correlation between nNOS and CD (r = −0.32; *p* = 0.04) with 11% of CD were related to nNOS (r^2^ = 0.11). A moderate positive statistically significant correlation was observed between VEGF and CD (r = 0.38; *p* = 0.03), and 14% of CD were dependent on VEGF expression (r^2^ = 0.14). APLNR and nNOS exhibited a strong positive significant correlation (r = 0.92; *p* < 0.001), with 84% of APLNR variability linked to nNOS (r^2^ = 0.84). Both APLNR (r = −0.58, r^2^ = 0.27) and nNOS (r = −0.39, r^2^ = 15) had moderate negative correlations with VEGF, which were statistically significant (*p* < 0.05).

In the RM of 12-month-old SHRs, a strong negative correlation was observed between APLNR and CD (r = −0.63), statistically significance (*p* < 0.01), with approximately only 39% of the variability in CD might be explained by APLNR (r^2^ = 0.39). A weak negative correlation was found between nNOS and CD (r = −0.24), not statistically significant (*p* = 0.2), indicating that only about 6% of the variability in CD can be explained by nNOS (r^2^ = 0.06). In contrast, a moderate negative correlation was observed between VEGF and CD (r = 0.43), statistically significant (*p* = 0.02), explaining approximately 18% of the variability in CD (r^2^ = 0.18). In addition, a moderate negative correlation was observed between APLNR and nNOS (r = −0.53), statistically significant (*p* = 0.01), with approximately 25% of the variability in nNOS explained by APLNR (r^2^ = 0.25). Similarly, a moderate negative correlation was found between APLNR and VEGF (r = −0.46), although it did not reach statistical significance (*p* = 0.01), with approximately 20% of the variability in VEGF explained by APLNR (r^2^ = 0.20). A strong positive correlation was revealed between nNOS and VEGF (r = 0.73), statistically significant (*p* < 0.001), suggesting a robust relationship between the two variables, with approximately 46% of the variability in VEGF explained by nNOS (r^2^ = 0.46). In the RM of age-matched controls we reported a weak negative statistically insignificant correlation between APLNR and CD (r = −0.09; *p* = 0.6), and only 2% of CD could be explained by APLNR expression (r^2^ = 0.02). On the other hand, nNOS and CD had a weak negative correlation which was not statistically significant (r = −0.27; *p* = 0.1), with only 8% of CD related to nNOS expression (r^2^ = 0.08). VEGF and CD had a moderate positive, marginally non-significant correlation (r = 0.34; *p* = 0.06), and 12% of CD was related to VEGF (r^2^ = 0.12). APLNR and nNOS had a moderate negative significant correlation (r = −0.48; *p* = 0.001), and 21% of APLNR expression could be explained by nNOS (r^2^ = 0.21). Similarly, a moderate negative correlation was observed between APLNR and VEGF (r = −0.37, *p* = 0.004) with about 14% of VEGF expression that could be explained by APLNR (r^2^ = 0.14). Lastly, a weak negative correlation was recorded between nNOS and VEGF (r = −0.08, *p* = 0.7, and r^2^ = 0.01). The results of the correlation analysis are summarized in Table 6 and Table 7, and graphically presented in Figure 6.

## 4. Discussion

The precise localization of APLNR within kidney structures remains unclear. Our study is one of the few reports in the literature on APLNR immunohistochemical expression in the kidney. We demonstrated that APLNR expression in the RC was predominantly localized on the cell membrane of epithelial cells in proximal and distal tubules, with increased expression in 12-month-old SHRs compared to 6-month-old SHRs as well as to age-matched controls. In the RM, APLNR expression was observed in the collecting duct epithelium, with a similar trend of increased expression in older SHRs compared to age-matched controls. According to other immunohistochemistry studies, APLNR is distributed across all segments of the nephron, with high expression levels noted in podocytes, vascular smooth muscle of renal arteries, and endothelial cells [17,49,50]. It has to be stated that apelin demonstrates protective effects against both acute and chronic kidney injuries. In animal models and in vitro ischemia/hypoxia–reperfusion models, apelin has shown renoprotective properties [51,52]. Its administration not only reduced tubular damage but also suppressed apoptotic pathways and diminished the expression of transforming growth factor-beta 1 (TGF-β1) [52]. Furthermore, apelin’s vasodilatory effects may offer protection against hypoxia by counteracting angiotensin II-induced vasoconstriction in the afferent and efferent glomerular arterioles [15] and enhancing renal medullary blood flow. This enhancement could potentially improve the perfusion of the outer medulla, which is particularly vulnerable to hypoxic injury [15,19,53]. Apelin has demonstrated protective effects in various forms of CKD. Studies conducted in rats with LNAME-induced hypertension or unilateral renal artery obstruction showed a decrease in both apelin and APLNR levels in cortical and medullar kidney extracts [17,54]. These findings are consistent with the observed downregulation of the renal apelin/APLNR pathway in diabetic animals [49,50], likely attributable to increased activity of the RAS. Even though they quantified both apelin and APLNR levels with blotting techniques, and we used only immunohistochemical semi-quantitative analysis, our results of upregulated APLNR are in direct contrast with these results. In general, there is presently no direct evidence to validate the concept that alterations in apelin levels influence the expression of APLNR in the kidneys. Xu et al. documented an increase in APLNR expression in the aorta and smooth muscle cells of SHRs while observing a simultaneous reduction in apelin levels [23]. Conversely, Sekerci noted a decline in apelin levels and APLNR downregulation in kidney tissues [17]. Consequently, further investigation is warranted to establish a potential correlation between APLNR expression and the circulating levels of its ligands. However, if we assume, based on the overwhelming amount of evidence in the literature, that apelin and elabela have a renoprotective function, then one logical explanation for the observed upregulation of APLNR would be due to the depletion of its ligands. Therefore, based on our results, we can propose that the apelin/APLNR system plays a key role in the sophisticated compensatory mechanisms occurring during the development and progression of AH, and disruptions in the apelinergic system might trigger AH deterioration. To test this hypothesis, we compared APLNR’s expression in SHRs to WRs and reported a statistically significant difference both RC and RM of 6- and 12-month-old SHRs compared to age-matched controls. To further elaborate on this claim, we correlated APLNR’s expression with CD and discovered an interesting trait. The correlation was negative in both RC and RM of young SHRs, and in both structures was weak and statistically not significant. However, with the progression of AH, we observed a strong negative correlation between APLNR immunoreactivity and CD in both RC and RM. In comparison in both the RC and RM of 6-month-old WRs, we recorded a moderate negative correlation between APLNR and CD, which was statistically significant. However, with age progression, the correlation between APLNR and CD in both RC and RM of 12-month-old WRs became weak negative one, with no statistical significance. With other words, in young SHRs, we report overall increased APLNR expression and capillary rarefication compared to age-matched controls. Furthermore, the upregulation of APLNR was recorded highest in the RC and RM of 12-month-old SHRs, where we also recorded the highest reduction in capillary numbers. On the other hand, the lowest expression of APLNR was recorded in the RM of 6-month-old WRs, where we recorded the highest number of capillaries. Moreover, in both RC and RM of old WRs, we again observed upregulation of APLNR and rarefication of capillaries with age progression, yet it was nowhere near as drastic as the one recorded with the progression and deterioration of AH. These data suggest that the upregulation of APLNR is connected with the deterioration of the vascularization of the kidney with AH progression. One plausible explanation would be that APLNR’s upregulation is in response to the depletion of apelin/elabela, as suggested by Xu et al. [23].

It has been suggested that apelin treatment lowers blood pressure in hypertensive models, further contributing to renal protection [54]. Moreover, apelin was found to be upregulated in a model of CKD induced by unilateral ureteral obstruction [55]. This research also proposed that the Akt/eNOS pathway could potentially facilitate some of the advantageous effects of apelin, and that the administration of losartan resulted in elevated levels of apelin and eNOS expression. [55], which would suggest synergism between the apelin/APLNR and NO/NOS systems. To further explore this supposed synergism between the two systems, we compared APLNR’s expression with the expression of nNOS in two different stages of AH and hypertension-induced kidney injury—at the onset of AH (6-month-old SHRs) and in advanced AH (12-month-old SHRs)—and compared it to age-matched WRs as controls. The expression of nNOS in the RC was primarily confined to the cellular membrane in both tubules and glomeruli, with older SHRs displaying higher expression compared to younger counterparts. That tendency was also observed in the controls. The expression of nNOS was significantly higher in both the RC and RM of 12-month-old SHRs compared to their age-matched controls. In the RM, nNOS expression was mainly reported in the cytoplasm of collecting duct epithelial cells, with older SHRs exhibiting higher expression. The immunohistochemical localization of nNOS in the kidney has been documented across different segments of the nephron, encompassing the macula densa, proximal tubule, the loop of Henle, distal tubule, and collecting duct [56]. Our immunohistochemical assessment revealed similar staining patterns between the expression of APLNR and nNOS in RC and RM of both age groups of SHRs and normotensive WRs. We further tested for differences in the expression of APLNR and nNOS with the post-hoc test. Significant differences between the expression of the two molecules were observed in the RC of 6-month-old SHRs, the RC and RM of 12-month-old SHRs, and in the RM of 12-month-old WRs. The reported discrepancy between the immunoreactivity of APLNR and nNOS is observed mainly in SHRs and in the structures with highest reduction of capillaries with AH progression. The results of the correlation analysis showed negative correlation between nNOS and CD across all studied structures in both age groups of SHRs and WRs, with the exception of the RC of 12-month-old SHRs, where we recorded a positive correlation between CD and nNOS. One potential explanation for this perplexing result might be the fact that in the RC of old SHRs, we observed the lowest amount of CD and the highest expression of nNOS, which further supports our hypothesis of nNOS involvement in the vascularizing compensatory mechanisms. Our hypothesis is further supported by the recorded difference in nNOS expression in the RC of older WRs and higher CD compared to the one in the RC of 12-month-old SHRs. The correlation analysis revealed a positive correlation between APLNR and nNOS in the RC and RM of young SHRs, as the correlation was proven statistically significant in the RM. These findings were also observed in the control group, where the correlation was even stronger. In 12-month-old SHRs, we observed a moderate negative correlation between APLNR and nNOS in both the RC and RM. This result was in direct contrast to the recorded strong positive correlation between APLNR and nNOS in the RC age-matched controls. Nevertheless, in the RM of 12-month-old WRs, the correlation was negative. These results suggest a potential disruption in the synergism of the apelin/APLNR and NO/nNOS systems in the RC with the development and progression of AH.

Moreover, apelin has been associated with promoting neoangiogenesis in proliferative retinopathies [57,58] and restoring myocardial vascular density [59,60]. It is not surprising, therefore, that some studies have suggested a correlation between apelin and VEGF levels [57,59], although this relationship remains somewhat contentious [61]. VEGF plays a crucial role in the glomerulus by facilitating communication between podocytes and endothelial cells [2]. The disruption of the glomerular VEGF pathway or the overproduction of VEGF has the potential to damage the glomerular filtration barrier and podocytes. [2,62]. Consequently, apelin-mediated VEGF secretion and angiogenesis might contribute to glomerular injury in diabetic nephropathy at a certain stage [16]. Moreover, recent advancements in oncological therapies utilizing anti-VEGF medications have highlighted the correlation between angiogenesis inhibition and the progression of AH [40,42]. Additionally, studies have suggested a potential interaction between the apelinergic system and VEGF [63]. Azad et al. propose that apelin inactivation enhances anti-VEGF therapy [63]. These findings suggest a potential link between the apelin/APLNR system and VEGF; therefore, we compared the immunoreactivity of APLNR with that of VEGF. In the RC, VEGF staining was most prominent in the visceral layer of Bowman’s capsule and the epithelial cells of proximal and distal tubular segments and was stronger in younger hypertensive animals compared to age-matched controls. In the RM, VEGF expression was reported predominantly in the collecting ducts and, to a lesser extent, in the loops of Henle and was again more pronounced in 6-month-old SHRs compared to 6-month-old WRs. In the literature, VEGF’s expression is predominantly described in glomerular podocytes and thick ascending limbs of Henle’s loop in normal kidneys, primarily signaling through VEGFR-2 on glomerular endothelial cells [64]. While podocytes do not secrete VEGFR, they produce VEGF, which acts in a paracrine manner to maintain the glomerular filtration barrier [65,66]. The results of the present study showed a significant difference between the expression of VEGF and APLNR in both the RC and RM at the onset of hypertension-induced renal injury (6-month-old SHRs), which was also recorded in normotensive controls. Similar was the tendency during the progression of AH (12-month-old SHRs), although we recorded an inversion of the immunoreactivity. Initially, VEGF’s expression was significantly higher than the expression of APLNR in both the RC and RM of both 6-month-old SHRs and WRs. However, with the progression of AH, APLNR’s immunoreactivity became significantly higher in both the RC and RM of 12-month-old SHRs. Similar were the results in the RM of 12-month-old controls, where the expression of APLNR was again higher than the expression of VEGF. Interestingly, VEGF had higher expression than APLNR in the RC of 12-month-old WRs. This strongly indicates that the two molecules are involved in a sophisticated compensatory vascular mechanism, likely interacting with each other. All correlation between APLNR and VEGF in the RC and RM of both age groups of hypertensive and normotensive animals were found to be negative and statistically significant. Our results strongly suggest that there is an established connection between these two molecules in the context of a vascular depletion during the progression of hypertension-induced renal damage. In addition, when we take into consideration the already established relationship between the apelin system and VEGF, then it becomes apparent that APLNR’s upregulation is part of a complex regulatory mechanism aiming to prevent AH deterioration.

The altered expression of VEGF may serve as a trigger mechanism for the development of hypertension-induced renal damage. Advani et al. [67] found elevated VEGF expression in SHRs compared to normotensive controls, proposing a potential renoprotective role of VEGF in hypertensive conditions. Liu et al. [68] reported a potential renoprotective effect of induced VEGF-A expression in SHRs, resulting in reduced inflammatory cell infiltration in the tubulointerstitium and better-preserved morphology of the glomerular filtration barrier. The importance of VEGF in maintaining renal microvasculature integrity has also been highlighted by Dimke et al. [64], who observed decreased peritubular CD following selective VEGF deletion in renal tubular segments. Reduced renal VEGF has been associated with glomerular capillary loss and implicated in the development of glomerulosclerosis [69]. Moreover, VEGF-mediated hypertrophy of remaining functional glomeruli acts as a compensatory mechanism in the early stages of glomerulosclerosis. However, prolonged nephron injury leads to peritubular and glomerular capillary loss and reduced VEGF expression [70]. Peritubular capillary rarefaction, associated with hypertensive nephrosclerosis, correlates positively with tubulointerstitial injury severity [11], influenced by factors such as endothelial–tubular epithelial dysfunction and inflammatory responses [71]. VEGF application in a remnant kidney model preserved renal function, stimulated endothelial cell growth, and maintained glomerular capillary integrity [Kang]. In diabetic nephropathy, VEGF levels correlated with glomerular and peritubular capillary loss [39]. Our correlation analysis reviewed statistically significant positive correlations between VEGF and CD in the RC and RM of both age groups, as the strongest correlation was observed in the RM of 6-month-old SHRs, and the weakest in the RM of 12-month-old SHRs. In the controls, we report similar tendency to a lesser extent; however, in the RM 12-month-old WRs, the correlation was found to be borderline statistically not significant. On the other hand, while VEGF expression in the RM was strongest at the onset of hypertension-induced renal damage, it was lowest in the RM of 12-month-old SHRs, and it therefore appears that the compensatory mechanism with the progression of AH in the RM was not as potent as in the RC. This claim was further confirmed by the lower values expressed in the control groups.

NO synthesis by NOS in endothelial cells plays a vital role in vasodilation and angiogenesis [72]. NO and VEGF are closely intertwined, with NO possibly serving as a downstream effector of VEGF and vice versa [73,74]. NO also regulates VEGF expression through various mechanisms in different cell types [75]. In general, the precise manner in which NO influences VEGF remains uncertain. Within the kidney, studies have demonstrated that VEGF triggers the phosphorylation of eNOS through pathways involving IRS-1/PI3K/AKT and ERK, leading to the generation of NO [76]. Our study revealed that initially, at the onset of AH, nNOS expression was low, while that of VEGF was significantly higher, both in young SHRs and WRs. However, with the progression of AH, nNOS expression matched that of VEGF, something only present in SHRs, as in the RC and RM of 12-month-old controls, we reported significant difference between the expression of nNOS and VEGF. Moreover, VEGF immunoreactivity was significantly higher in the RC of young animals and diminished with the deterioration of AH. The same depletion of VEGF was noted in the control group, yet not as potent. Conversely, nNOS expression, was lower in the RC of younger SHRs and gradually increased with AH progression. A similar tendency occurred in the RM of SHRs and in the age-matched control group, albeit less pronounced. In the RM of 6-month-old SHRs, VEGF expression was the strongest and weakened in the RM of 12-month-old SHRs. This age-related depletion of VEGF was also present in the controls, but not as prominent as in the SHR group. On the other hand, nNOS immunoreactivity was lowest in the RM of 6-month-old SHRs and increased in the RM of 12-month-old SHRs. Similar, but less pronounced findings were documented in controls. The post-hoc test also confirmed this tendency, as it revealed a statistically significant difference between the expression of nNOS and VEGF in the RC and RM of young SHRs and WRs, and a non-significant difference between these two molecules in the RC and RM of old SHRs and only in the medulla of WRs. These results suggest that the elevation of nNOS expression with AH progression is perhaps an attempt to promote VEGF secretion as a compensatory mechanism. The observed correlations between nNOS and VEGF in both RC and RM of both age groups of experimental animals were similar to the correlation between APLNR and VEGF. The only difference being that the correlations between nNOS and VEGF were positive only in the RC and RM of 12-month-old SHRs. Similar findings were recorded in controls; however, all correlations between nNOS and VEGF were negative. The inversion of the correlation between nNOS and VEGF from negative in the RC and RM of 6-month-old SHRs to positive in 12-month-old SHRs strongly implies the synergism of this two molecules during the progression of AH. This hypothesis is further supported by the lack of such inversion of the correlations between nNOS and VEGF in the age matched controls.

Based on the overwhelming body of evidence presented in the current study, as well as the preexisting literature data, we can hypothesize that the complex interplay of APLNR, nNOS, and VEGF is a key component of a sophisticated vascular adaptive mechanism occurring during the progression of hypertension-induced renal damage. We therefore suggest that at the onset of AH, VEGF expression is high in an attempt to maintain adequate renal vascularization, correlating with high CD and lower expression of APLNR and nNOS. However, with the progression of AH and associated renal injury, VEGF expression is depleted, which correlates with a significant decrease in CD and stronger immunoreactivity of APLNR and nNOS as a secondary part of this compensatory mechanism, aiming to activate additional molecular cascades in order to stimulate VEGF production and restore adequate renal perfusion. Nonetheless, further investigation is required to comprehend the complex molecular interactions, which are at play during the pathogenesis and deterioration of AH.

The current study acknowledges several limitations. Firstly, we exclusively used immunohistochemical methods to visualize the molecules, and we employed a semi-quantitative approach to evaluate their expression in the kidney. It is important to recognize the inherent variability when visually quantifying immunohistochemistry slides among different observers. To mitigate this issue, we used an automated software program, as demonstrated in previous research [77]. Secondly, we focused solely on male SHRs to minimize potential confounding effects associated with female sex hormones and periodic fluctuations observed in female SHRs. Thirdly, it is worth noting that the rat kidney has a species-specific characteristic of being unipapillar, with only one pyramid, which may limit the applicability of our findings to the human population. Nevertheless, our work paves the way for future research on understanding the complex pathogenesis of hypertension-induced renal injury and potentially for the development of better treatment.

## 5. Conclusions

The present study explored the immunohistochemical localization of APLNR, nNOS, and VEGF in the kidney of SHRs, as well as their potential interactions in the context of hypertension-induced renal damage. Our results demonstrate a significant decrease in renal CD, alongside significant depletion in VEGF levels with AH progression. Our study strongly suggests that APLNR, nNOS, and VEGF are essential components of a sophisticated compensatory vascular mechanism aiming to restore VEGF levels and adequate renal perfusion in order to mitigate hypertension-induced renal damage. Overall, our findings provide valuable insight into the intricate interplay between APLNR, nNOS, and VEGF in the pathogenesis of hypertension-induced renal injury. Further research is warranted to elucidate the precise mechanisms underlying these interactions and their potential therapeutic implications in renal pathology.

## Figures and Tables

**Figure 1 biomedicines-12-01723-f001:**
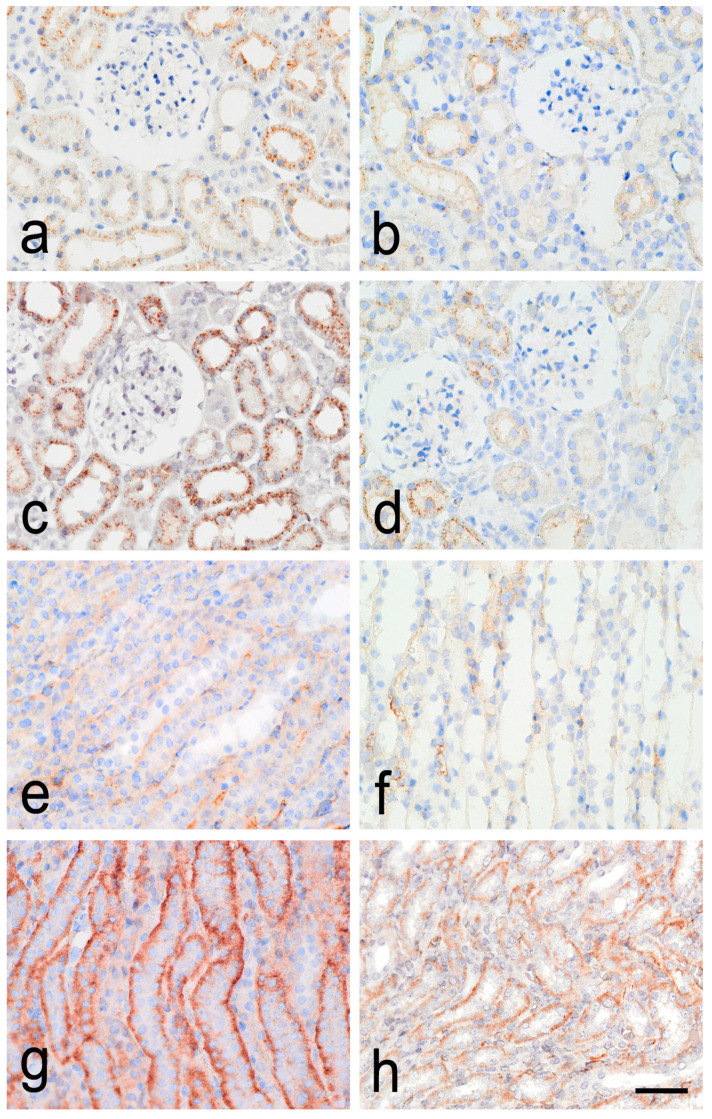
Immunohistochemical staining for apelin receptor (APLNR) in the kidney of spontaneously hypertensive rats (SHRs) and normotensive Wistar rats (WRs). (**a**)—renal cortex (RC) of 6-month-old SHR; (**b**)—renal cortex (RC) of 6-month-old WR; (**c**)—renal cortex (RC) of 12-month-old SHR; (**d**)—renal cortex (RC) of 12-month-old WR; (**e**)—renal medulla (RM) of 6-month-old SHR; (**f**)—renal medulla (RM) of 6-month-old WR; (**g**)—renal medulla (RM) of 12-month-old SHR; (**h**)—renal medulla (RM) of 12-month-old WR; scale bar 25 μm.

**Figure 2 biomedicines-12-01723-f002:**
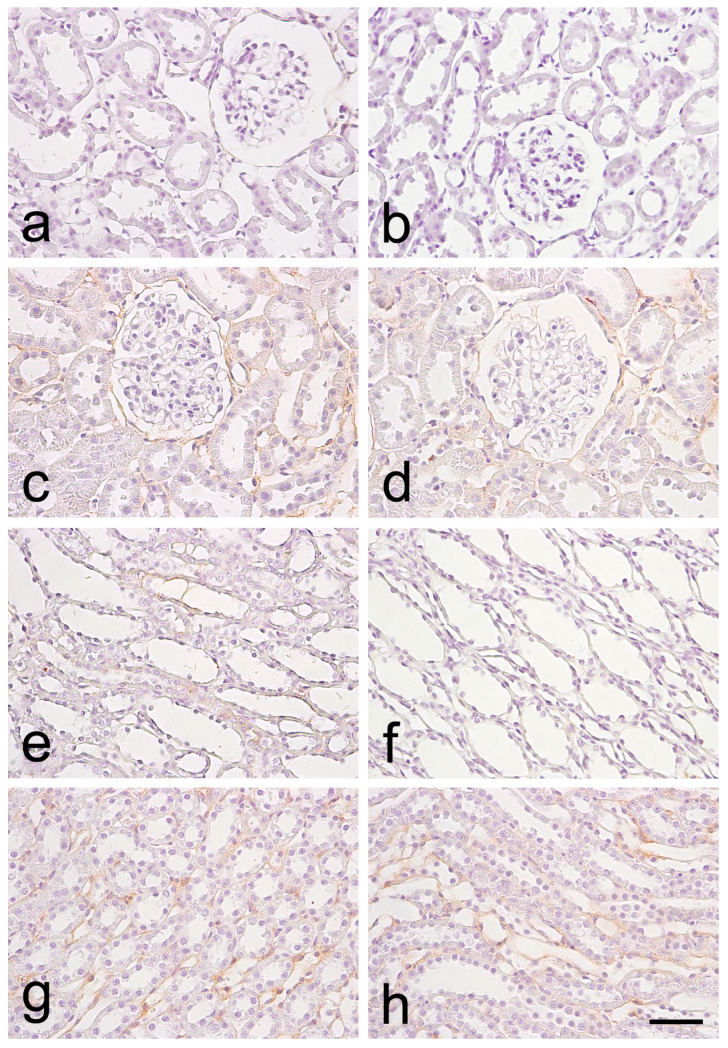
Immunohistochemical staining for neuronal nitric oxide synthase (nNOS) in the kidney of spontaneously hypertensive rats (SHRs) and normotensive Wistar rats (WRs). (**a**)—renal cortex (RC) of 6-month-old SHR; (**b**)—renal cortex (RC) of 6-month-old WR; (**c**)—renal cortex (RC) of 12-month-old SHR; (**d**)—renal cortex (RC) of 12-month-old WR; (**e**)—renal medulla (RM) of 6-month-old SHR; (**f**)—renal medulla (RM) of 6-month-old WR; (**g**)—renal medulla (RM) of 12-month-old SHR; (**h**)—renal medulla (RM) of 12-month-old WR; scale bar 25 μm.

**Figure 3 biomedicines-12-01723-f003:**
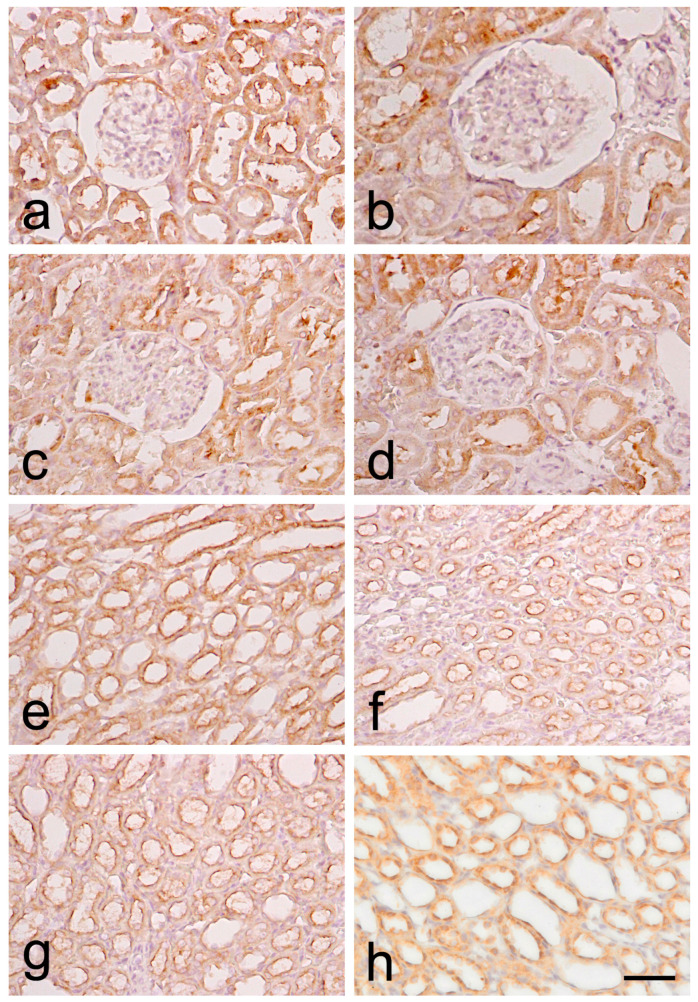
Immunohistochemical staining for vascular endothelial growth factor (VEGF) in the kidney of spontaneously hypertensive rats (SHRs) and normotensive Wistar rats (WRs). (**a**)—renal cortex (RC) of 6-month-old SHR; (**b**)—renal cortex (RC) of 6-month-old WR; (**c**)—renal cortex (RC) of 12-month-old SHR; (**d**)—renal cortex (RC) of 12-month-old WR; (**e**)—renal medulla (RM) of 6-month-old SHR; (**f**)—renal medulla (RM) of 6-month-old WR; (**g**)—renal medulla (RM) of 12-month-old SHR; (**h**)—renal medulla (RM) of 12-month-old WR; scale bar 25 μm.

**Figure 4 biomedicines-12-01723-f004:**
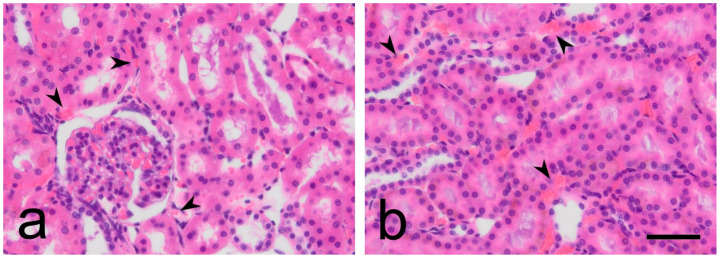
Hematoxylin and eosin staining of kidney. (**a**)—cortex; (**b**)—medulla. Black arrowhead—capillaries. Scale bar—25 μm.

**Figure 5 biomedicines-12-01723-f005:**
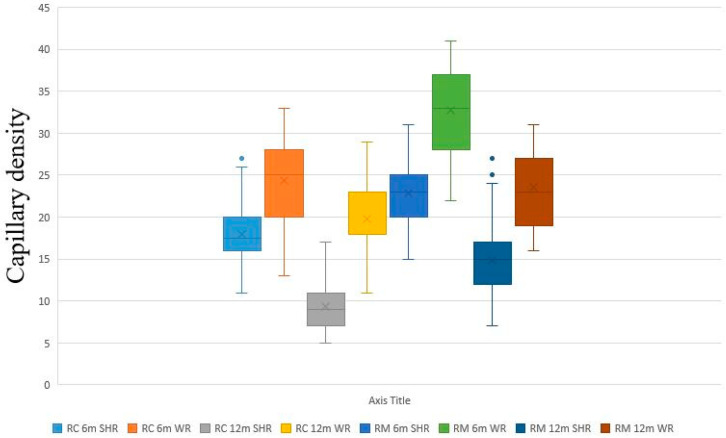
Graphical representation of the parameter capillary density in the kidney of spontaneously hypertensive rats (SHRs) and normotensive Wistar rats (WRs) as controls presented with box and whisker plot showing the median (square), surrounded by a ‘box’, the vertical edge of which is the interval between the lower and upper quartile [25–75%]. ‘Whiskers’ originating from this ‘box’ represent the non-outlier range. Circles—outliers. RC—renal cortex; RM—renal medulla.

**Figure 6 biomedicines-12-01723-f006:**
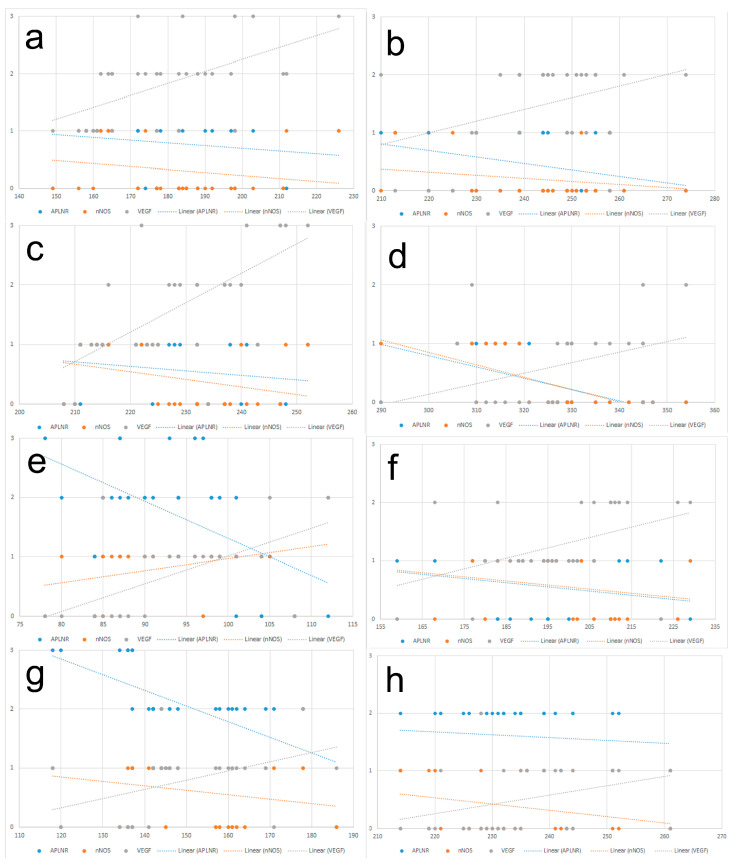
Graphical representation of correlations between capillary density (CD) and the semi-quantitative assessment of the expression of apelin receptor (APLNR), neuronal nitric oxide synthase (nNOS), and vascular endothelial growth factor (VEGF) in 6- and 12-month-old spontaneously hypertensive rats (SHRs) and normotensive Wistar rats (WRs) as controls. (**a**)—renal cortex (RC) of 6-month-old SHR; (**b**)—renal cortex (RC) of 6-month-old WR; (**c**)—renal cortex (RC) of 12-month-old SHR; (**d**)—renal cortex (RC) of 12-month-old WR; (**e**)—renal medulla (RM) of 6-month-old SHR; (**f**)—renal medulla (RM) of 6-month-old WR; (**g**)—renal medulla (RM) of 12-month-old SHR; (**h**)—renal medulla (RM) of 12-month-old WR; x-axis—number of capillaries; y-axis—semi-quantitative assessment of the expression of the apelin receptor (APLNR)—blue dots; neuronal nitric oxide synthase (nNOS)—orange dots; vascular endothelial growth factor (VEGF)—grey dots.

**Table 1 biomedicines-12-01723-t001:** Mean systolic (mmHg) and mean diastolic blood pressure (mmHg) of 6- and 12-month-old spontaneously hypertensive rats and normotensive Wistar rats. Each group consisted of six animals (n = 6). SHRs—spontaneously hypertensive rats; WRs—Wistar rats; SD—standard deviation.

Age Group	Mean Systolic Blood Pressure (mmHg) ± SD	Mean Diastolic Blood Pressure (mmHg) ± SD
6-month-old SHRs	174.7 ± 2.6	115.9 ± 2.8
6-month-old WRs	118 ± 1.9	77 ± 2.2
12-month-old SHRs	201.8 ± 3.7	125.8 ± 1.9
12-month-old WRs	122 ± 2.1	83 ± 3.1

**Table 2 biomedicines-12-01723-t002:** Semi-quantitative analysis of the intensity of immunohistochemical staining for the apelin receptor, neuronal nitric oxide synthase, and vascular endothelial growth factor in the renal cortex and renal medulla in 6- and 12-month-old spontaneously hypertensive rats (SHRs) and normotensive Wistar rats (WRs). The analysis was performed with IHC Profiler. APLNR—apelin receptor; nNOS—neuronal nitric oxide synthase; RC—renal cortex; RM—renal medulla; VEGF—vascular endothelial growth factor.

Experimental Animals		APLNR	nNOS	VEGF
6-month-old rats	RC of SHRs	High-positive (3+) (0%) Positive (2+) (3%) Low-positive (1+) (68%) Negative (0) (29%)	High-positive (3+) (0%) Positive (2+) (0%) Low-positive (1+) (34%) Negative (0) (66%)	High-positive (3+) (20%) Positive (2+) (42%) Low-positive (1+) (37%) Negative (0) (1%)
	RC of WRs	High-positive (3+) (0%) Positive (2+) (0%) Low-positive (1+) (42%) Negative (0) (58%)	High-positive (3+) (0%) Positive (2+) (0%) Low-positive (1+) (19%) Negative (0) (81%)	High-positive (3+) (0%) Positive (2+) (67%) Low-positive (1+) (19%) Negative (0) (14%)
	RM of SHRs	High-positive (3+) (0%) Positive (2+) (0%) Low-positive (1+) (56%) Negative (0) (44%)	High-positive (3+) (0%) Positive (2+) (0%) Low-positive (1+) (39%) Negative (0) (61%)	High-positive (3+) (29%) Positive (2+) (40%) Low-positive (1+) (26%) Negative (0) (5%)
	RM of WRs	High-positive (3+) (0%) Positive (2+) (0%) Low-positive (1+) (28%) Negative (0) (72%)	High-positive (3+) (0%) Positive (2+) (0%) Low-positive (1+) (26%) Negative (0) (74%)	High-positive (3+) (0%) Positive (2+) (10%) Low-positive (1+) (45%) Negative (0) (45%)
12-month-old rats	RC of SHRs	High-positive (3+) (12%) Positive (2+) (53%) Low-positive (1+) (17%) Negative (0) (18%)	High-positive (3+) (0%) Positive (2+) (4%) Low-positive (1+) (76%) Negative (0) (20%)	High-positive (3+) (0%) Positive (2+) (9%) Low-positive (1+) (51%) Negative (0) (40%)
	RC of WRs	High-positive (3+) (0%) Positive (2+) (0%) Low-positive (1+) (54%) Negative (0) (46%)	High-positive (3+) (0%) Positive (2+) (0%) Low-positive (1+) (56%) Negative (0) (44%)	High-positive (3+) (0%) Positive (2+) (33%) Low-positive (1+) (55%) Negative (0) (12%)
	RM of SHRs	High-positive (3+) (23%) Positive (2+) (66%) Low-positive (1+) (7%) Negative (0) (4%)	High-positive (3+) (0%) Positive (2+) (0%) Low-positive (1+) (62%) Negative (0) (38%)	High-positive (3+) (0%) Positive (2+) (7%) Low-positive (1+) (63%) Negative (0) (30%)
	RM of WRs	High-positive (3+) (0%) Positive (2+) (70%) Low-positive (1+) (21%) Negative (0) (9%)	High-positive (3+) (0%) Positive (2+) (0%) Low-positive (1+) (38%) Negative (0) (62%)	High-positive (3+) (0%) Positive (2+) (3%) Low-positive (1+) (54%) Negative (0) (43%)

**Table 3 biomedicines-12-01723-t003:** Analysis of significance in the difference of expression of the apelin receptor (APLNR), neuronal nitric oxide synthase (nNOS), and vascular endothelial growth factor (VEGF) assessed semi-quantitatively in the kidney of 6- and 12-month-old spontaneously hypertensive rats (SHRs) compared to age-matched normotensive Wistar rats (WRs) as controls. APLNR-C—expression of apelin receptor in the renal cortex; APLNR-M—expression of apelin receptor in the renal medulla; nNOS-C—expression of neuronal nitric oxide synthase in the renal cortex; nNOS-M—expression of neuronal nitric oxide synthase in the renal medulla; VEGF-C—expression of vascular endothelial growth factor in the renal cortex; VEGF-M—expression of vascular endothelial growth factor in the renal medulla; *p*—*p*-value.

Compared Groups	*p*-Value
APLNR-C 6m SHR/APLNR-C 6m WR	0.07
nNOS-C 6m SHR/nNOS-C 6m WR	0.25
VEGF-C 6m SHR/VEGF-C 6m WR	0.04
APLNR-M 6m SHR/APLNR-M 6m WR	0.02
nNOS-M 6m SHR/nNOS-M 6m WR	0.28
VEGF-M 6m SHR/VEGF-M 6m WR	0.001
APLNR-C 12m SHR/APLNR-C 12m WR	0.001
nNOS-C 12m SHR/nNOS-C 12m WR	0.04
VEGF-C 12m SHR/VEGF-C 12m WR	0.001
APLNR-M 12m SHR/APLNR-M 12m WR	0.03
nNOS-M 12m SHR/nNOS-M 12m WR	0.04
VEGF-M 12m SHR/VEGF-M 12m WR	0.04

**Table 4 biomedicines-12-01723-t004:** Descriptive statistics for the parameter capillary density per high-power field on hematoxylin and eosin-stained slides in the kidney of 6- and 12-month-old spontaneously hypertensive rats and controls. RC—renal cortex; RM—renal medulla; SHRs—spontaneously hypertensive rats; WRs—Wistar rats; SD—standard deviation.

Experimental Animals		Mean	SD	Median	Min	Max	*p*-Value
6-month-old rats	RC SHRs	18.0	3.1	18.5	16	27	*p* < 0.05
	RC WRs	24.3	4.8	25	13	33	
	RM SHRs	22.9	3.8	22.5	20	31	*p* < 0.05
	RM WRs	32.8	5.1	28	22	41	
12-month-old rats	RC SHRs	9.3	2.8	7.5	6	17	*p* < 0.05
	RC WRs	19.8	4.6	28.5	11	29	
	RM SHRs	14.8	3.8	11	7	27	*p* < 0.05
	RM WRs	23.5	4.2	19.5	16	31	

**Table 5 biomedicines-12-01723-t005:** Analysis of significance in the difference of expression of the apelin receptor (APLNR), neuronal nitric oxide synthase (nNOS), and vascular endothelial growth factor (VEGF) assessed semi-quantitatively in the kidney of 6- and 12-month-old spontaneously hypertensive rats (SHRs) as well as age-matched normotensive Wistar rats (WRs) as controls. APLNR-C—expression of apelin receptor in the renal cortex; APLNR-M—expression of apelin receptor in the renal medulla; nNOS-C—expression of neuronal nitric oxide synthase in the renal cortex; nNOS-M—expression of neuronal nitric oxide synthase in the renal medulla; VEGF-C—expression of vascular endothelial growth factor in the renal cortex; VEGF-M—expression of vascular endothelial growth factor in the renal medulla; *p*—*p*-value.

Compared Groups	*p*-Value
APLNR-C 6m SHR/nNOS-C 6m SHR	0.005
APLNR-C 6m WR/nNOS-C 6m WR	0.3
APLNR-C 6m SHR/VEGF-C 6m SHR	0.001
APLNR-C 6m WR/VEGF-C 6m WR	0.001
nNOS-C 6m SHR/VEGF-C 6m SHR	0.001
nNOS-C 6m WR/VEGF-C 6m WR	0.001
APLNR-M 6m SHR/nNOS-M 6m SHR	0.6
APLNR-M 6m WR/nNOS-M 6m WR	0.96
APLNR-M 6m SHR/VEGF-M 6m SHR	0.001
APLNR-M 6m WR/VEGF-M 6m WR	0.02
nNOS-M 6m SHR/VEGF-M 6m SHR	0.001
nNOS-M 6m WR/VEGF-M 6m WR	0.02
APLNR-C 12m SHR/nNOS-C 12m SHR	0.001
APLNR-C 12m WR/nNOS-C 12m WR	0.9
APLNR-C 12m SHR/VEGF-C 12m SHR	0.001
APLNR-C 12m WR/VEGF-C 12m WR	0.001
nNOS-C 12m SHR/VEGF-C 12m SHR	0.8
nNOS-C 12m WR/VEGF-C 12m WR	0.001
APLNR-M 12m SHR/nNOS-M 12m SHR	0.001
APLNR-M 12m WR/nNOS-M 12m WR	0.001
APLNR-M 12m SHR/VEGF-M 12m SHR	0.001
APLNR-M 12m WR/VEGF-M 12m WR	0.001
nNOS-M 12m SHR/VEGF-M 12m SHR	0.6
nNOS-M 12m WR/VEGF-M 12m WR	0.7

**Table 6 biomedicines-12-01723-t006:** Descriptive correlation analysis between capillary density (CD) and the semi-quantitative assessment of the expression of the apelin receptor (APLNR), neuronal nitric oxide synthase (nNOS), and vascular endothelial growth factor (VEGF) in the kidney of 6- and 12-month-old spontaneously hypertensive rats (SHRs) and age-matched normotensive Wistar rats (WRs) as controls. CD-C—cortical capillary density; CD-M—medullary capillary density; APLNR-C—expression of apelin receptor in the renal cortex; APLNR-M—expression of apelin receptor in the renal medulla; nNOS-C—expression of neuronal nitric oxide synthase in the renal cortex; nNOS-M—expression of neuronal nitric oxide synthase in the renal medulla; VEGF-C—expression of vascular endothelial growth factor in the renal cortex; VEGF-M—expression of vascular endothelial growth factor in the renal medulla; r—Pearson correlation coefficient; r^2^—coefficient of determination; *p*—*p*-value.

Correlated Parameters	r	r^2^	*p*-Value
CD-C 6m SHR/APLNR-C 6m SHR	−0.19	0.04	0.3
CD-C 6m WR/APLNR-C 6m WR	−0.35	0.13	0.04
CD-M 6m SHR/APLNR-M 6m SHR	−0.24	0.05	0.3
CD-M 6m WR/APLNR-M 6m WR	−0.31	0.1	0.04
CD-C 6m SHR/nNOS-C 6m SHR	−0.25	0.06	0.2
CD-C 6m WR/nNOS-C 6m WR	−0.32	0.11	0.04
CD-M 6m SHR/nNOS-M 6m SHR	−0.31	0.09	0.04
CD-M 6m WR/nNOS-M 6m WR	−0.33	0.12	0.07
CD-C 6m SHR/VEGF-C 6m SHR	0.58	0.34	0.0008
CD-C 6m WR/VEGF-C 6m WR	0.39	0.15	0.03
CD-M 6m SHR/VEGF-M 6m SHR	0.66	0.43	0.00004
CD-M 6m WR/VEGF-M 6m WR	0.38	0.14	0.03
CD-C 12m SHR/APLNR-C 12m SHR	−0.54	0.29	0.002
CD-C 12m WR/APLNR-C 12m WR	−0.13	0.02	0.5
CD-M 12m SHR/APLNR-M 12m SHR	−0.63	0.39	0.00005
CD-M 12m WR/APLNR-M 12m WR	−0.09	0.02	0.6
CD-C 12m SHR/nNOS-C 12m SHR	0.38	0.15	0.04
CD-C 12m WR/nNOS-C 12m WR	−0.22	0.05	0.2
CD-M 12m SHR/nNOS-M 12m SHR	−0.24	0.06	0.2
CD-M 12m WR/nNOS-M 12m WR	−0.27	0.08	0.1
CD-C 12m SHR/VEGF-C 12m SHR	0.61	0.38	0.002
CD-C 12m WR/VEGF-C 12m WR	0.45	0.20	0.01
CD-M 12m SHR/VEGF-M 12m SHR	0.43	0.18	0.02
CD-M 12m WR/VEGF-M 12m WR	0.34	0.12	0.06

**Table 7 biomedicines-12-01723-t007:** Descriptive correlation analysis between the semi-quantitative assessment of the expression of the apelin receptor (APLNR), neuronal nitric oxide synthase (nNOS), and vascular endothelial growth factor (VEGF) in the kidney of 6- and 12-month-old spontaneously hypertensive rats (SHRs) and age-matched normotensive Wistar rats (WRs) as controls. APLNR-C—expression of apelin receptor in the renal cortex; APLNR-M—expression of apelin receptor in the renal medulla; nNOS-C—expression of neuronal nitric oxide synthase in the renal cortex; nNOS-M—expression of neuronal nitric oxide synthase in the renal medulla; VEGF-C—expression of vascular endothelial growth factor in the renal cortex; VEGF-M—expression of vascular endothelial growth factor in the renal medulla; r—Pearson correlation coefficient; r^2^—coefficient of determination; *p*—*p*-value.

Correlated Parameters	r	r^2^	*p*-Value
APLNR-C 6m SHR/nNOS-C 6m SHR	−0.18	0.09	0.3
APLNR-C 6m WR/nNOS-C 6m WR	0.22	0.07	0.4
APLNR-C 6m SHR/VEGF-C 6m SHR	−0.31	0.10	0.04
APLNR-C 6m WR/VEGF-C 6m WR	−0.44	0.21	0.01
nNOS-C 6m SHR/VEGF-C 6m SHR	−0.38	0.15	0.03
nNOS-C 6m WR/VEGF-C 6m WR	−0.42	0.19	0.02
APLNR-M 6m SHR/nNOS-M 6m SHR	0.69	0.43	<0.001
APLNR-M 6m WR/nNOS-M 6m WR	0.92	0.84	<0.001
APLNR-M 6m SHR/VEGF-M 6m SHR	−0.54	0.26	0.01
APLNR-M 6m WR/VEGF-M 6m WR	−0.58	0.27	0.01
nNOS-M 6m SHR/VEGF-M 6m SHR	−0.42	0.18	0.02
nNOS-M 6m WR/VEGF-M 6m WR	−0.39	0.15	0.02
APLNR-C 12m SHR/nNOS-C 12m SHR	−0.48	0.21	0.01
APLNR-C 12m WR/nNOS-C 12m WR	0.93	0.88	<0.001
APLNR-C 12m SHR/VEGF-C 12m SHR	−0.41	0.18	0.02
APLNR-C 12m WR/VEGF-C 12m WR	−0.32	0.11	0.04
nNOS-C 12m SHR/VEGF-C 12m SHR	0.36	0.13	0.04
nNOS-C 12m WR/VEGF-C 12m WR	−0.30	0.10	0.04
APLNR-M 12m SHR/nNOS-M 12m SHR	−0.53	0.25	0.01
APLNR-M 12m WR/nNOS-M 12m WR	−0.48	0.21	0.01
APLNR-M 12m SHR/VEGF-M 12m SHR	−0.46	0.20	0.01
APLNR-M 12m WR/VEGF-M 12m WR	−0.37	0.14	0.004
nNOS-M 12m SHR/VEGF-M 12m SHR	0.73	0.46	<0.001
nNOS-M 12m WR/VEGF-M 12m WR	−0.08	0.01	0.7

## Data Availability

The raw data supporting the conclusions of this article will be made available by the authors upon request.

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
