# Peer review of "Unraveling the Complex Molecular Interplay and Vascular Adaptive Changes in Hypertension-Induced Kidney Disease"

_biomedicines, 2024, doi:10.3390/biomedicines12081723_

Round 1

Reviewer 1 Report (Previous Reviewer 1)

Comments and Suggestions for Authors

The authors have revised the manuscript as suggested. I have no comments on the current content. I request publication of the paper.

Author Response

Response to reviewer comments

Reviewer 1:
The authors have revised the manuscript as suggested. I have no comments on the current content. I request publication of the paper.

Reviewer 2:
I do not have further comments.

Reviewer 3:
The authors have addressed the reviewer's comments.

Reviewer 4:
In the study, the aurthors investigate the role of apelin receptor (APLNR), neuronal nitric oxide synthase (nNOS), and vascular endothelial growth factor (VEGF) in arterial hypertension-induced renal damage. The results showed significant differences in molecule expression between age groups and varying correlations between the expression of these molecules and capillary density.

The paper is well written, but min issues should be addressed to improve the paper.

  1. The authors should inserting the following paper in the introduction: doi: 10.3390/ijms21155294. PMID: 32722551; PMCID: PMC7432634.
    We appreciate your suggestion. Indeed, the recommended paper is of excellent quality and provides in-depth information about the role of VEGF-A in cardiomyocytes and heart diseases. However, in our view, this paper is not directly related to the primary focus of our work—the kidney. Therefore, we have chosen not to include it in the introduction, as it predominantly addresses cardiomyocytes and heart conditions such as ischemic heart disease and atherosclerosis. We have previously cited this paper multiple times in our earlier works regarding the expression of VEGF in the myocardium.
  2. The authors should explain why they did not perform a power analysis for the number of animals used in association with the experimental activity performed.
    In this experimental study, we used the same number of animals as in several previously published studies by our team (Stanchev et al., 2023; Stanchev et al., 2020; Kotov et al., 2020). The number of animals was also selected based on earlier studies by other researchers who used a similar number of models. To enhance the statistical power of our study, we examined multiple fields per slide. We have adhered to internationally recognized guidelines and the practices adopted by our institution regarding the humane treatment of experimental animals and the conservation of animal resources. Furthermore, a post hoc power analysis performed using G*Power software version 3.1.9.7 indicated that a sample size of six rats per group would yield a power of 0.82 with an alpha level of 0.05. Additionally, our study utilizes 300 vision fields, further enhancing the statistical power.

Reviewer 2 Report (Previous Reviewer 2)

Comments and Suggestions for Authors

I do not have further comments.

Author Response

Response to reviewer comments

Reviewer 1:
The authors have revised the manuscript as suggested. I have no comments on the current content. I request publication of the paper.

Reviewer 2:
I do not have further comments.

Reviewer 3:
The authors have addressed the reviewer's comments.

Reviewer 4:
In the study, the aurthors investigate the role of apelin receptor (APLNR), neuronal nitric oxide synthase (nNOS), and vascular endothelial growth factor (VEGF) in arterial hypertension-induced renal damage. The results showed significant differences in molecule expression between age groups and varying correlations between the expression of these molecules and capillary density.

The paper is well written, but min issues should be addressed to improve the paper.

  1. The authors should inserting the following paper in the introduction: doi: 10.3390/ijms21155294. PMID: 32722551; PMCID: PMC7432634.
    We appreciate your suggestion. Indeed, the recommended paper is of excellent quality and provides in-depth information about the role of VEGF-A in cardiomyocytes and heart diseases. However, in our view, this paper is not directly related to the primary focus of our work—the kidney. Therefore, we have chosen not to include it in the introduction, as it predominantly addresses cardiomyocytes and heart conditions such as ischemic heart disease and atherosclerosis. We have previously cited this paper multiple times in our earlier works regarding the expression of VEGF in the myocardium.
  2. The authors should explain why they did not perform a power analysis for the number of animals used in association with the experimental activity performed.
    In this experimental study, we used the same number of animals as in several previously published studies by our team (Stanchev et al., 2023; Stanchev et al., 2020; Kotov et al., 2020). The number of animals was also selected based on earlier studies by other researchers who used a similar number of models. To enhance the statistical power of our study, we examined multiple fields per slide. We have adhered to internationally recognized guidelines and the practices adopted by our institution regarding the humane treatment of experimental animals and the conservation of animal resources. Furthermore, a post hoc power analysis performed using G*Power software version 3.1.9.7 indicated that a sample size of six rats per group would yield a power of 0.82 with an alpha level of 0.05. Additionally, our study utilizes 300 vision fields, further enhancing the statistical power.

Reviewer 3 Report (Previous Reviewer 3)

Comments and Suggestions for Authors

The authors have addressed the reviewer's comments. 

Author Response

Response to reviewer comments

Reviewer 1:
The authors have revised the manuscript as suggested. I have no comments on the current content. I request publication of the paper.

Reviewer 2:
I do not have further comments.

Reviewer 3:
The authors have addressed the reviewer's comments.

Reviewer 4:
In the study, the aurthors investigate the role of apelin receptor (APLNR), neuronal nitric oxide synthase (nNOS), and vascular endothelial growth factor (VEGF) in arterial hypertension-induced renal damage. The results showed significant differences in molecule expression between age groups and varying correlations between the expression of these molecules and capillary density.

The paper is well written, but min issues should be addressed to improve the paper.

  1. The authors should inserting the following paper in the introduction: doi: 10.3390/ijms21155294. PMID: 32722551; PMCID: PMC7432634.
    We appreciate your suggestion. Indeed, the recommended paper is of excellent quality and provides in-depth information about the role of VEGF-A in cardiomyocytes and heart diseases. However, in our view, this paper is not directly related to the primary focus of our work—the kidney. Therefore, we have chosen not to include it in the introduction, as it predominantly addresses cardiomyocytes and heart conditions such as ischemic heart disease and atherosclerosis. We have previously cited this paper multiple times in our earlier works regarding the expression of VEGF in the myocardium.
  2. The authors should explain why they did not perform a power analysis for the number of animals used in association with the experimental activity performed.
    In this experimental study, we used the same number of animals as in several previously published studies by our team (Stanchev et al., 2023; Stanchev et al., 2020; Kotov et al., 2020). The number of animals was also selected based on earlier studies by other researchers who used a similar number of models. To enhance the statistical power of our study, we examined multiple fields per slide. We have adhered to internationally recognized guidelines and the practices adopted by our institution regarding the humane treatment of experimental animals and the conservation of animal resources. Furthermore, a post hoc power analysis performed using G*Power software version 3.1.9.7 indicated that a sample size of six rats per group would yield a power of 0.82 with an alpha level of 0.05. Additionally, our study utilizes 300 vision fields, further enhancing the statistical power.

Reviewer 4 Report (New Reviewer)

Comments and Suggestions for Authors

In the study, the aurthors investigate the role of apelin receptor (APLNR), neuronal nitric oxide synthase (nNOS), and vascular endothelial growth factor (VEGF) in arterial hypertension-induced renal damage. The results showed significant differences in molecule expression between age groups and varying correlations between the expression of these molecules and capillary density.

The paper is well written, but min issues should be addressed to improve the paper.

1. The authors should inserting the following paper in the introduction: doi: 10.3390/ijms21155294. PMID: 32722551; PMCID: PMC7432634.

2. The authors should explain why they did not perform a power analysis for the number of animals used in association with the experimental activity performed.

Author Response

Response to reviewer comments

Reviewer 1:
The authors have revised the manuscript as suggested. I have no comments on the current content. I request publication of the paper.

Reviewer 2:
I do not have further comments.

Reviewer 3:
The authors have addressed the reviewer's comments.

Reviewer 4:
In the study, the aurthors investigate the role of apelin receptor (APLNR), neuronal nitric oxide synthase (nNOS), and vascular endothelial growth factor (VEGF) in arterial hypertension-induced renal damage. The results showed significant differences in molecule expression between age groups and varying correlations between the expression of these molecules and capillary density.

The paper is well written, but min issues should be addressed to improve the paper.

  1. The authors should inserting the following paper in the introduction: doi: 10.3390/ijms21155294. PMID: 32722551; PMCID: PMC7432634.
    We appreciate your suggestion. Indeed, the recommended paper is of excellent quality and provides in-depth information about the role of VEGF-A in cardiomyocytes and heart diseases. However, in our view, this paper is not directly related to the primary focus of our work—the kidney. Therefore, we have chosen not to include it in the introduction, as it predominantly addresses cardiomyocytes and heart conditions such as ischemic heart disease and atherosclerosis. We have previously cited this paper multiple times in our earlier works regarding the expression of VEGF in the myocardium.
  2. The authors should explain why they did not perform a power analysis for the number of animals used in association with the experimental activity performed.
    In this experimental study, we used the same number of animals as in several previously published studies by our team (Stanchev et al., 2023; Stanchev et al., 2020; Kotov et al., 2020). The number of animals was also selected based on earlier studies by other researchers who used a similar number of models. To enhance the statistical power of our study, we examined multiple fields per slide. We have adhered to internationally recognized guidelines and the practices adopted by our institution regarding the humane treatment of experimental animals and the conservation of animal resources. Furthermore, a post hoc power analysis performed using G*Power software version 3.1.9.7 indicated that a sample size of six rats per group would yield a power of 0.82 with an alpha level of 0.05. Additionally, our study utilizes 300 vision fields, further enhancing the statistical power.

This manuscript is a resubmission of an earlier submission. The following is a list of the peer review reports and author responses from that submission.

Round 1

Reviewer 1 Report

Comments and Suggestions for Authors

Congratulations to the authors on a very interesting idea.  The background of the hypothesis is given in a very comprehensive and fluent physiological introduction. 

I have a couple of suggestions

1/ The authors describe what is in the tables, and they also take the time to point out correlations that are not statistically significant. This is not necessary, in my opinion, and the description of the results should have been more concise.

2/ The p-value is easier to read if given in hundreds.

3/ A critical, interesting discussion of the results should be shorter.

4/ There are redundancies, such as the nephroprotective effect of apelin and elabel.

5/ Line 56- Word not translated -[Addendum to reference].

Author Response

1/ The authors describe what is in the tables, and they also take the time to point out correlations that are not statistically significant. This is not necessary, in my opinion, and the description of the results should have been more concise.
- we agree with the suggestion and have rewriten our results section in accordance

2/ The p-value is easier to read if given in hundreds.
- we agree and have given them in hundreds in our revised manuscript

3/ A critical, interesting discussion of the results should be shorter.
- we appreciate the suggestion, however, since we included normotensive controls in our revised manuscript, we were not able to shorten it much.

4/ There are redundancies, such as the nephroprotective effect of apelin and elabel.
- we appreciate the suggestion, yet we are using nephroprotective effect of apelin and elabela as an indirect marker for the renoprotective effects of APLNR.

5/ Line 56- Word not translated -[Addendum to reference].
- has been corrected in the revised version

Reviewer 2 Report

Comments and Suggestions for Authors

Gaydarski et al investigates capillary density and expression of APLNR, nNOS and VEGF in spontaneous hypertensive rats. The study is of potential interest; however, this study is highly descriptive with a lack of proper controls and important additional measurements.

For example:

The experiments in SHR must be controlled with normotensive WKY.

The work is based on immunohistological images and semi quantitative analysis. While this method is helpful, other molecular biology methods such as western blot and qPCR must be used to assess the targets of interest.

Capillary density in kidneys can be demonstrated by tissue staining methods which authors must perform and present.

Some additional suggestions:

What about sex differences? Investigation of male and female rats would be interesting.

If APLNR is regulating angiogenesis, this could be tested in a simple cell culture method, e.g. endothelial ell sprouting.

Comments on the Quality of English Language

English language is largely acceptable.

Author Response

The experiments in SHR must be controlled with normotensive WKY.
_ we agree with the suggestion and have included age-matched normotensive Wistar rats as controls in our revised version.

The work is based on immunohistological images and semi quantitative analysis. While this method is helpful, other molecular biology methods such as western blot and qPCR must be used to assess the targets of interest.
- we appreciate your suggestion, however, such studies have already been conducted for all of the three molecules we assess, yet no other study have previously compared the simultaneous immunohistochemical expression of APLNR, nNOS and VEGF in the renal cortex and medulla. Furthermore, we evaluate the cahnges in the expresiion of the target molecules in two age groups (6-month-old - onset of hypertension and 12-month-old - developed hypertension) and compare them to the expression in the age-matched controls.

Capillary density in kidneys can be demonstrated by tissue staining methods which authors must perform and present.
- we accept your suggestion and have included Hematoxylin and Eosin stained slides of the renal cortex and medulla in order to demonstrate the capillaries

Some additional suggestions:

What about sex differences? Investigation of male and female rats would be interesting.
-we appreciate your suggestion, yet we have pointed this out as one of our limitations, as well as the reasoning behind it - "we focused solely on male animals to minimize potential confounding effects associated with female sex hormones and periodic fluctuations observed in female animals."

If APLNR is regulating angiogenesis, this could be tested in a simple cell culture method, e.g. endothelial ell sprouting.
- we appreciate your suggestion, yet we are aiming for in-vivo evidences about the complex vasculary mechanisms involved in the pathogenesis of arterial hypertension.

Reviewer 3 Report

Comments and Suggestions for Authors

1. The experimental design could not clarify the causal relationship between vascular adaptive mechanisms in hypertension-induced kidney disease.

2. The quality of immunohistochemistry was low which cannot support the conclusion of this study.

3. Immunohistochemistry is not an adequate methodology for quantifying the expression of proteins. 

Comments on the Quality of English Language

 Minor editing of English language required

Author Response

1. The experimental design could not clarify the causal relationship between vascular adaptive mechanisms in hypertension-induced kidney disease.
-_ we appreciate your concern, therefore, we have included age-matched normotensive Wistar rats as controls in our revised version.

2. The quality of immunohistochemistry was low which cannot support the conclusion of this study.
- we appreciate your comment and have improoved the quality of the figures.

3. Immunohistochemistry is not an adequate methodology for quantifying the expression of proteins. 
-we appreciate your suggestion, however, we aim to compar the simultaneous immunohistochemical expression of APLNR, nNOS and VEGF in the renal cortex and medulla. Furthermore, we evaluate the cahnges in the expresiion of the target molecules in two age groups (6-month-old - onset of hypertension and 12-month-old - developed hypertension) and compare them to the expression in the age-matched controls.

Round 2

Reviewer 2 Report

Comments and Suggestions for Authors

The authors only partly addressed my comments. The major issue still remaining in this work is highly descriptive nature. This is not adequate of the title 'Unraveling the Complex Vascular Adaptive Mechanisms in Hypertension-Induced Kidney Disease'. There is nothing mechanistic about this work, especially the complex mechanisms of vascular remodeling in the kidneys. 

Mechanistic studies are needed to adequately address the title of this work.

The images of Figures 1 and 2 seem identical to me. So it is heard to understand where the differences proposed by the authors are.

Author Response

The authors only partly addressed my comments. The major issue still remaining in this work is highly descriptive nature. This is not adequate of the title 'Unraveling the Complex Vascular Adaptive Mechanisms in Hypertension-Induced Kidney Disease'. There is nothing mechanistic about this work, especially the complex mechanisms of vascular remodeling in the kidneys. Mechanistic studies are needed to adequately address the title of this work.
- we appreciate the comment and have changed the title of our work, as we agree our work is more focused on the morphological changes in the expression of the APLNR, nNOS and VEGF during hypertension progression. The main goal of our work is the display and compare the morphological localization of the examined molecules. Moreover, we utilized capillary density as a quantitative measure of hypertension-induced kidney damage and correlated it with the expression of the APLNR, nNOS and VEGF.

The images of Figures 1 and 2 seem identical to me. So it is heard to understand where the differences proposed by the authors are.
- we appreciate your concern, therefore in order to comply with the requirements, we have redone the immunohistochemistry and rewritten extensively our results and discussion section.

Reviewer 3 Report

Comments and Suggestions for Authors

The authors have addressed the reviewer's comments.

The style of references needs to be revised.

Author Response

The authors have addressed the reviewer's comments.

The style of references needs to be revised.
- we have revised the references in accordance with the journal requirements.